# Nitrogen Fertilizer and Nitrapyrin for Greenhouse Gas Reduction in Wolfberry Orchards on the Qinghai–Tibetan Plateau

Jiujin Lu [1,†], Yunzhang Xu [1,2,†], Haiyan Sheng [1,2,*], Yajun Gao [2,3], Jim Moir [4] , Rong Zhang [5] and Shouzhong Xie [6]

1   Department of Agriculture and Forestry, College of Agriculture and Animal Husbandry, Qinghai University, Xining 810016, China; 1990990013@qhu.edu.cn (J.L.); 2021990071@qhu.edu.cn (Y.X.)
2   State Key Laboratory of Plateau Ecology and Agriculture, Qinghai University, Xining 810016, China; yajungao@nwsuaf.edu.cn
3   College of Natural Resources and Environment, Northwest A&F University, Xinyang 712100, China
4   Soil Science Department, Lincoln University, Lincoln, 7647 Christchurch, New Zealand; jim.moir@lincoln.ac.nz
5   Qinghai Academy of Agriculture and Forestry, Xining 810016, China; 1997990031@qhu.edu.cn
6   Nuomuhong Farm, Dulan 816100, China; 1998990011@qhu.edu.cn
*   Correspondence: 1990990011@qhu.edu.cn
†   These authors contributed equally to this work.

**Abstract:** Wolfberry production has become a major agro-industry on the Qinghai–Tibetan Plateau, causing increased nitrogen (N) pollution and greenhouse gas (GHG) emissions. Appropriate N fertilizer rate and nitrification inhibitors may mitigate GHG emissions and improve N use efficiency. A 2-year field experiment was conducted to measure the effects of N application rate and nitrapyrin on GHG emissions, to reduce GHG emissions and N pollution. We used eight treatments: Control (CK), 667 kg·ha$^{-1}$ N (Con), 400 kg·ha$^{-1}$ N (N$_{400}$), 267 kg·ha$^{-1}$ N (N$_{267}$), 133 kg·ha$^{-1}$ N (N$_{133}$), 400 kg·ha$^{-1}$ N plus 2.00 kg·ha$^{-1}$ nitrapyrin (N$_{400}$I$_{2.00}$), 267 kg·ha$^{-1}$ N plus 1.33 kg·ha$^{-1}$ nitrapyrin (N$_{267}$I$_{1.33}$) and 133 kg·ha$^{-1}$ N plus 0.67 kg·ha$^{-1}$ nitrapyrin (N$_{133}$I$_{0.67}$). Compared with Con treatment, N$_{400}$ maintained fruit yield and increased net income, but saved 40% of N fertilizer and decreased the cumulative N$_2$O emission by 14–16%. Compared to N$_{400}$, N$_{267}$ and N$_{133}$ treatments, the cumulative N$_2$O emission of N$_{400}$I$_{2.00}$, N$_{267}$I$_{1.33}$ and N$_{133}$I$_{0.67}$ treatments was reduced by 28.5–45.1%, 26.6–29.9% and 33.8–45.9%, respectively. Furthermore, N$_{400}$I$_{2.00}$ resulted in the highest wolfberry yield and net income. The emissions of CH$_4$ and CO$_2$ were not significantly different among treatments. Moreover, the global warming potential (GWP) and the greenhouse gas emission intensity (GHGI) of N$_{400}$I$_{2.00}$ declined by 45.6% and 48.6% compared to Con treatment. Therefore, 400 kg·ha$^{-1}$ N combined with 2.00 kg·ha$^{-1}$ nitrapyrin was shown to be a promising management technique for maintaining wolfberry yield while minimizing GWP and GHGI.

**Keywords:** wolfberry; N fertilizer rate; nitrapyrin; greenhouse gas emissions; yield

## 1. Introduction

Global warming due to excessive GHG emissions is now a serious global issue drawing unprecedented scientific and political attention. China is one of the largest emitters of GHG. The country aims to decrease GHG by 60–65% per unit of gross domestic product by 2030 [1]. Agriculture, the largest anthropogenic source, directly releases considerable amounts of GHG to the atmosphere [2,3]; this accounts for 17% of total release in the world, but close to 30% in China [4]. On a 100-year scale, the global warming potential of CH$_4$ and N$_2$O is 28 and 265 times than that of CO$_2$, respectively [5]. It is predicted that annual GHG emissions in arable land will be about 564 Tg CO$_2$-equivalent in 2030 [6].

To meet the growing demand for food, N fertilizer input is increasing in agriculture, especially in China. Wolfberry (*Lycium barbarum* L.), a salt-tolerant and drought-resistant



shrub, is mainly grown in northwest China [7,8]. Because of the unique climatic condition and geographical environment, wolfberry production has become a major agro-industry in Qaidam, on the Qinghai–Tibetan Plateau. Plantings reached 49,900 ha in 2020 [9]. However, in the search for high yields, excessive N fertilizer input in wolfberry production has reduced N use efficiency (NUE) and increased costs, N pollution and GHG emissions [10–12]. Moreover, previous studies have shown that high N inputs negatively affect soluble solids and other quality indicators of wolfberry fruit [13–15]. Therefore, rational N fertilization is of great significance to provide a theoretical basis for reducing GHG emission and improving wolfberry yield and quality.

Optimizing N fertilizer applications and using nitrification inhibitors may be effective strategies to improve N use rate and reduce the environmental costs of agricultural production [16,17]. It was revealed that $N_2O$ emission significantly decreased due to reduced N fertilizer rates [18–21]. The cumulative $N_2O$ emission decreased by 68.8% during the experiment when N fertilizer was decreased by 53.3% [22]. Wang et al. [23] found that the $N_2O$ loss was lowered by 72.2 g·ha$^{-1}$ when fertilizer N input was 33.3% lower. In apple orchards, when N fertilizer was decreased from 800–400 kg·ha$^{-1}$ N, $N_2O$ cumulative emission decreased by 43.3% [24]. On the basis of standard current farmer practice, N application rate decreased by 22.9% and 21.9%, resulting in the reduction of GHGI [25]. A reduction in N application rate also helped to decrease $CH_4$ emissions [26].

Nitrapyrin can delay the conversion of ammonium ($NH_4^+$) to nitrate ($NO_3^-$) by inhibiting the activities of nitrifying bacteria in soil. This improves N retention in the soil and enhances N utilization rates [27,28]. Zhou et al. [29] found that nitrapyrin could decrease $N_2O$ emissions and GHGI by 61.4% and 28.5%, respectively, in paddy fields. In the Midwest of the United States, applying nitrapyrin significantly lowered GHG emissions, while the yield and soil N retention increased by 7.0% and 28.0%, respectively [30]. Moreover, a nitrification inhibitor effectively undermined the denitrification in an apple orchard by 15.8–53.1% [31]. It is worth noting that there is a low residual amount of the nitrification inhibitor itself in the soil, and there is no ecological risk [32,33]. Consequently, nitrapyrin and optimum N fertilizer rate have broad potential in mitigating GHG emissions and preventing environmental pollution.

At present, excessive N fertilizer significantly increases GHG emissions, which has negative effects on wolfberry quality and the environment. In the Qaidam area, the application of N fertilizer in production of wolfberry lacks scientific theoretical basis, and the problems of low N utilization rate, waste of agricultural resources and high production cost caused by blindly investing a large amount of N fertilizer are common. Furthermore, few reports have focused on the effects of optimum N fertilizer rate and nitrapyrin on GHG emissions in wolfberry on the Tibetan Plateau. Therefore, we hypothesized that reasonable N fertilizer rate and nitrapyrin, in combination, could mitigate greenhouse gas emissions. A 2-year field study was carried out in a wolfberry orchard on the Qinghai-Tibetan Plateau to: (i) investigate greenhouse gas fluxes across optimum N fertilizer rate combined with nitrapyrin; (ii) elucidate the effects of optimum N fertilizer rate combined with nitrapyrin on GHG emissions and wolfberry yield; (iii) quantify GWP and GHGI of $CH_4$ and $N_2O$ emission under optimum N fertilizer rate combined with nitrapyrin.

## 2. Materials and Methods

### 2.1. Description of Study Site

A field experiment was conducted at Nomuhong Farm (96°20′ E, 36°25′ N) in Golmud City, Qinghai Province, Qinghai–Tibet Plateau from 2019 to 2020, with an altitude of 2760 masl and a typical plateau continental climate. The average annual sunshine duration is 3600 h and the evaporation is 2800–3000 mm per annum. About 66.8% of the annual precipitation occurs from June to October. The soil type is sandy loam classified as grey-brown desert soil according to Chinese Soil Taxonomy [11]. The physicochemical properties of 0–20 cm soil were determined according to Bao [34], and the main parameters were: 1.51 g·cm$^{-3}$ bulk density, 8.49 pH, 1.43 g·kg$^{-1}$ total N, 3.05 g·kg$^{-1}$ total P, 23.1 g·kg$^{-1}$ total K,

19.5 g·kg$^{-1}$ organic matter, 82.6 mg·kg$^{-1}$ Olsen-P, 69.76 mg·kg$^{-1}$ alkaline hydrolyzed nitrogen and 210.8 mg·kg$^{-1}$ available K. Wolfberry is the predominant economic plant in Nuomuhong farm with an increasing area planted per annum.

## 2.2. Experimental Design and Field Management

An area of 10-year-old wolfberry trees (*Lycium barbarum* L.) was selected for the experimental site; the experiments were carried out in the same field for two consecutive years during 2019 and 2020. A randomized complete block design was employed with 8 treatments (3 replications). The treatments were: (i) no N fertilization (CK), (ii) 667 kg·ha$^{-1}$ N (Con), (iii) 400 kg·ha$^{-1}$ N (N$_{400}$), (iv) 267 kg·ha$^{-1}$ N (N$_{267}$), (v) 133 kg·ha$^{-1}$ N (N$_{133}$), (vi) 400 kg·ha$^{-1}$ N plus 2.00 kg·ha$^{-1}$ nitrapyrin (N$_{400}$I$_{2.00}$), (vii) 267 kg·ha$^{-1}$ N plus 1.33 kg·ha$^{-1}$ nitrapyrin (N$_{267}$I$_{1.33}$), (viii) 133 kg·ha$^{-1}$ N plus 0.67 kg·ha$^{-1}$ nitrapyrin (N$_{133}$I$_{0.67}$). Each plot was 39 m$^2$ (13 m $\times$ 3 m), with a row spacing of 2 m and plant spacing within rows of 1.5 m.

Commercial organic fertilizer (Qinghai Enze Agricultural Technology Co., Ltd., Xining, China; the fertilizer is the product of sheep manure fermentation after maturity: organic matter, 51.7%; N, 3.7%; P$_2$O$_5$, 1.1%; K$_2$O, 2.0%) was applied at 1667 kg·ha$^{-1}$ and calcium triple superphosphate (Yuntianhua Group Co., Ltd., Kunming, China, P$_2$O$_5$, 46%) at 724 kg·ha$^{-1}$ (equivalent to 333 kg·ha$^{-1}$ pure P$_2$O$_5$) were applied in each treatment, which is the typical rate used by farmers. Half of the total N (Urea, Yuntianhua Group Co., Ltd., Kunming, China, N, 46%) and 50% nitrapyrin (Zhejiang Aofutol Chemical Co., Ltd., Shaoxing, China, 70%) input with commercial organic fertilizer and calcium triple superphosphate were applied as basal fertilizers, which were evenly distributed into fertilization holes (0.30 m away from root, 0.20 m deep $\times$ 0.50 m long $\times$ 0.25 m wide) between rows on 19 May 2019 and 15 May 2020. The remainder (50% total N and nitrapyrin) was top-dressed in the same location on 30 June 2019 and 5 July 2020. Urea and nitrapyrin of each treatment were mixed thoroughly before fertilization. The wolfberry orchard was flood irrigated 7 times a year with a total irrigation quota of 6000 m$^3$·ha$^{-1}$. Other field managements, such as irrigation, were the same as that used by local farmers.

## 2.3. Gas Sampling and Measurements

Greenhouse gas samples were collected in each plot from May to October in 2019 and 2020, using a closed static chamber–gas chromatography method [35]. Each gas collecting equipment included a chamber and a base. The chambers (0.5 m deep $\times$ 0.5 m long $\times$ 0.5 m wide) were composed of a stainless-steel frame, and covered with cystosepiment to prevent dramatic temperature changes inside the chamber. The chamber base was inserted 20 cm into the soil before the experiment. A three-way valve was connected at a distance of 35 cm from the bottom of the chamber, and a temperature detection port was arranged beside the valve. A small fan was installed at the top of the box's diagonal corner to mix gas. The chamber was fitted into the frame base with a groove (2 cm width) during sampling, and the groove was full of water to seal. Sampling was conducted from 9:00–12:00 am. At 0, 15, 30 and 45 min after the chamber was closed, the gas was extracted with a 50 mL syringe and injected into the sealed air bags. Gas samples were determined by gas chromatography (Agilent 7890A, Agilent Technologies, Palo Alto, USA) equipped with a $^{63}$Ni-electron capture detector for N$_2$O detection at 350 °C and a flame ionization detector for CH$_4$ and CO$_2$ detection at 200 °C. Frequency of gas collections was daily for 7 days after fertilization, 3 days after irrigation, 1 additional measurement at the time of daily precipitation more than 20 mm; the remaining experimental time samples were collected once a week [35].

## 2.4. Environmental Factors Measurements

Air temperature and precipitation data were obtained from a meteorological station adjacent to the experimental site. A thermometer was used to measure the 10 cm soil temperature before the first gas sample was collected and after the last gas sample was taken. A digital thermometer within the chamber provided the average air temperature

inside the chamber, which was used to calculate the gas flux. Wolfberry fruit from each plot was collected to determine the weight of fresh fruit on 29 July, 20 August and 14 September in 2019 and 4 August, 25 August and 19 September in 2020. The fresh fruits were dried in a greenhouse and then weighed.

*2.5. Calculation*

Soil gravimetric water content was calculated to 10 cm depth according to Equation (1). Gravimetric water content and soil bulk density were separately determined with the oven-drying method and cutting ring method [36].

$$\text{WFPS}(\%) = \frac{\text{Soil gravimetric water content}(\%)}{1 - \frac{\text{Soil bulk density}}{2.65}} \times 100\% \tag{1}$$

Fluxes of $N_2O$, $CH_4$ and $CO_2$ were estimated using Equation (2) [37].

$$F = \frac{273}{273 + T} \times \frac{M}{V} \times H \times \frac{d_c}{d_t} \tag{2}$$

where F ($\mu g\ N_2O\text{-}N\ m^{-2}\cdot h^{-1}$ or $\mu g\ CH_4\text{-}C\ m^{-2}\cdot h^{-1}$ or $mg\ CO_2\text{-}C\ m^{-2}\cdot h^{-1}$) is the net flux. In addittion, T (°C) is the mean of air temperature in the chamber, M (28 g $N_2O$-N $mol^{-1}$ for $N_2O$, 12 g $CH_4$-C $mol^{-1}$ for $CH_4$, 12 g $CO_2$-C $mol^{-1}$ for $CO_2$) is the molecular weight of $N_2$ or C. V is mole volume (22.4 L·$mol^{-1}$) at 273 K and 1013 hPa, H (m) is the chamber height, c is the concentration of $N_2O$ ($\mu L \cdot L^{-1}$), $CH_4$ ($\mu L \cdot L^{-1}$) or $CO_2$ ($mL \cdot L^{-1}$) in volume mixing ratio. Finally, t (h) is the time of chamber closure, dc/dt ($\mu L \cdot L^{-1} \cdot h^{-1}$ for $N_2O$ and $CH_4$, $mL \cdot L^{-1} \cdot h^{-1}$ for $CO_2$) is the initial rate of change in $N_2O$, $CH_4$ or $CO_2$ concentration in the chamber enclosure [38].

Gas fluxes and cumulative emissions were calculated using a linear or non-linear model [39,40] and the direct interpolation method [41,42], respectively. Direct $N_2O$ emission factor was calculated according to the following formula:

$$\text{EF}_{N_2O}(\%) = \frac{R - R_{CK}}{F} \times 100\% \tag{3}$$

where R and $R_{CK}$ separately represent the cumulative $N_2O$ emission (kg·$ha^{-1}$ $N_2O$-N) from applied N fertilizer per plot and CK plot, and F is the N fertilizer rate for one year (kg·$ha^{-1}$ N).

GWP is calculated using Equation (4) where the unit of GWP is $CO_2$-eq kg·$ha^{-1}$; $R_{CH4}$ and $R_{N2O}$ are the cumulative $CH_4$ and $N_2O$ emissions (kg·$ha^{-1}$). On the time scale of 100 years, the GWP of unit mass $CH_4$ and $N_2O$ is 265 times and 28 times of $CO_2$.

$$\text{GWP} = R_{CH_4} \times \frac{16}{12} \times 28 + R_{N_2O} \times \frac{44}{28} \times 265 \tag{4}$$

GHGI is an indicator for comprehensive evaluation of greenhouse effect. As shown in Equation (5), its unit is $CO_2$-eq kg·$Mg^{-1}$; the yield is in wolfberry dried fruits (Mg·$ha^{-1}$).

$$\text{GHGI} = \frac{\text{GWP}}{\text{Yield}} \tag{5}$$

*2.6. Statistical Analysis*

Using IBM SPSS Statistics 23.0, and all raw data were verified with respect to exhibiting normality (Kolmogorov–Smirnov test) and homogeneity of variance (Levene's test). For multiple comparisons, we used Fisher's protected least significant differences (LSD) at the 5% level of probability. Pearson's correlation was used to analyze the correlation between greenhouse gas emission and soil water-filled pore space (WFPS) or 10 cm soil temperature ($T_{soil}$). We performed linear, quadratic and exponential curve fittings to simulate the response of greenhouse gas emission to N rate, then used the determination coefficient ($R^2$)

to select the optimum curve. Origin 2018 software (OriginLab, Northampton, USA) was used for drawing figures.

## 3. Results

### 3.1. Environmental Factors

The total rainfall during the wolfberry growing season was 48.3 mm in 2019 (139 days) and 28.6 mm in 2020 (144 days); 13.7 mm precipitation occurred on 3 July 2019, accounting for 28.4% of the total precipitation (Figure 1). The air temperature increased to seasonal highs and then decreased during each observation period (from May to October). The maximum and minimum air temperatures in 2019 and 2020 during the growing season were 25.0 °C and 21.7 °C in August, 5.8 °C and 3.2 °C in October, respectively. The average air temperature was lower in 2020 (14.4 °C) than in 2019 (17.5 °C).

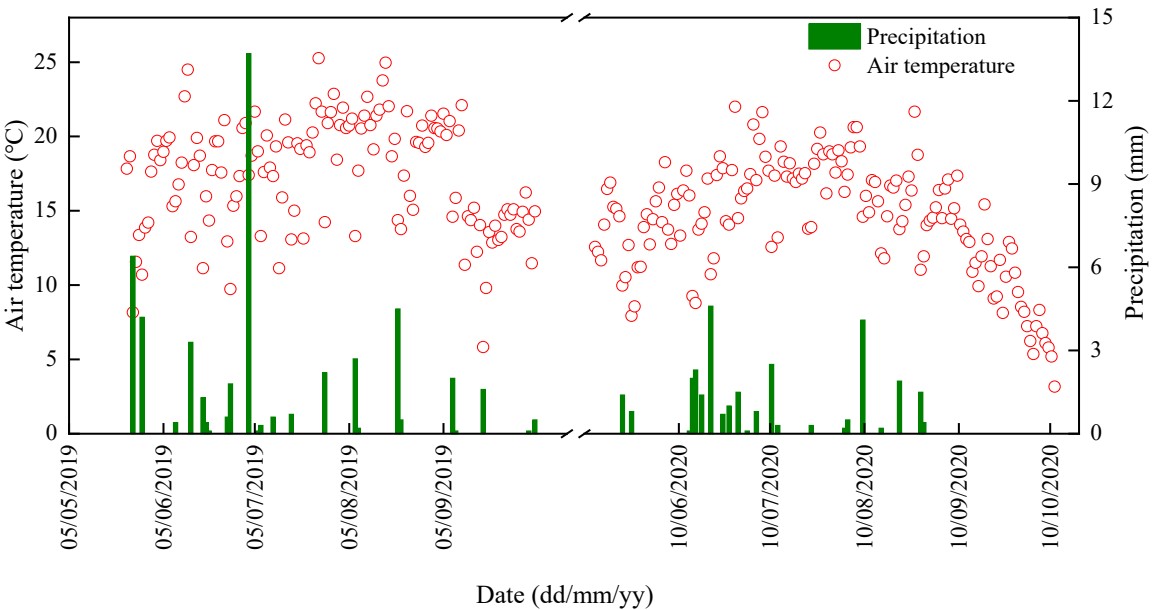

**Figure 1.** Precipitation and air temperature during experiment.

WFPS increased following seven flood irrigation events. Due to heavy rain on 3 July 2019 and 21 June 2020, WFPS peaked after the second irrigation in two years. There was no difference in average WFPS between 2019 (58.3%) and 2020 (58.2%), nor was there a significant difference in WFPS among treatments in the same year. The 10 cm soil temperature decreased and then increased with the occurrence of irrigation, reaching the highest value in August in both years (Figure 2). The peak values of the 10 cm soil temperature were 19.5 °C and 21.1 °C in 2019 and 2020, while the minimum was 8.1 °C and 5.6 °C, respectively. The mean value of the 10 cm soil temperature was 13.7 °C in 2019, with an increase of 6.2% over 2020.

### 3.2. GHG Emissions

#### 3.2.1. $N_2O$ Emission

The $N_2O$ emissions after N fertilizer application or irrigation significantly increased to various degrees (Figure 3a). Peak $N_2O$ emission was observed for 3–5 days after N fertilization following irrigation. The $N_2O$ peaks of Con treatment reached 1765.40 $\mu g \cdot m^{-2} \cdot h^{-1}$ on 19 May 2019 (fertilization combined with irrigation on 16 May 2019) and 1783.30 $\mu g \cdot m^{-2} \cdot h^{-1}$ on 8 July 2020 (top-dressing N fertilizer following irrigation on 5 July 2020). $N_{400}I_{2.00}$ resulted in lower $N_2O$ peaks than Con over the two years. Moreover, emission peaks of $N_{400}I_{2.00}$, $N_{267}I_{1.33}$ and $N_{133}I_{0.67}$ treatments were lower than that of $N_{400}$, $N_{267}$ and $N_{133}$ treatments. Pearson's correlation showed $N_2O$ emission was significantly positively correlated with WFPS and 10 cm soil temperature ($p < 0.05$) (Figure 4).

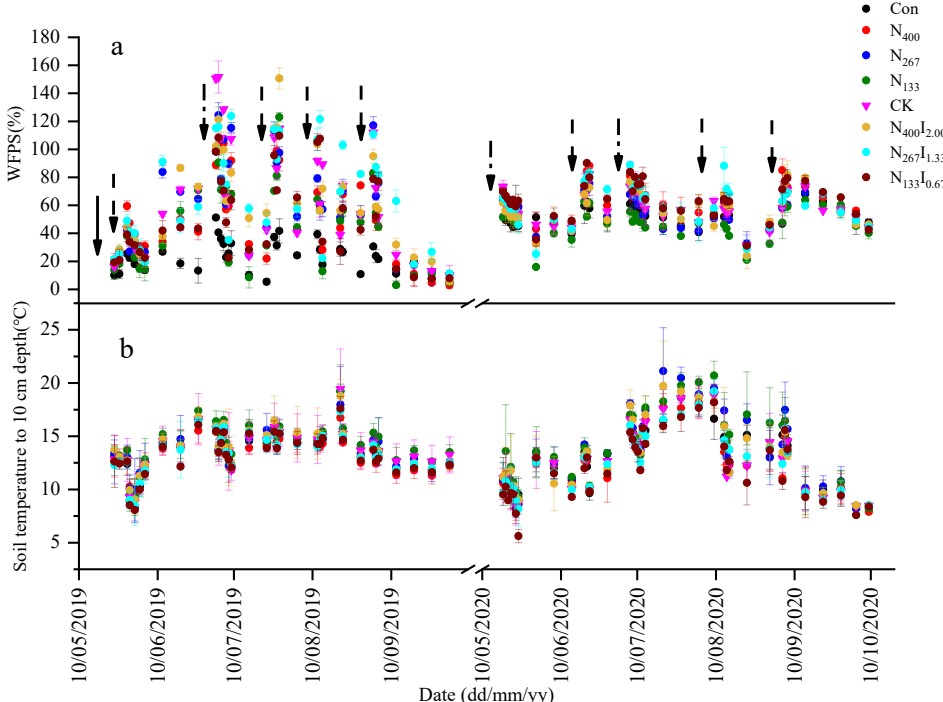

**Figure 2.** Dynamics of soil WFPS (**a**) and 10 cm soil temperature (**b**) during the experimental period. Black solid and black dashed arrows separately indicate fertilization and irrigation dates. Black dashed/dotted arrows represent the fertilizer date combined with irrigation. Vertical bars indicate standard error (*n* = 3), the same as below.

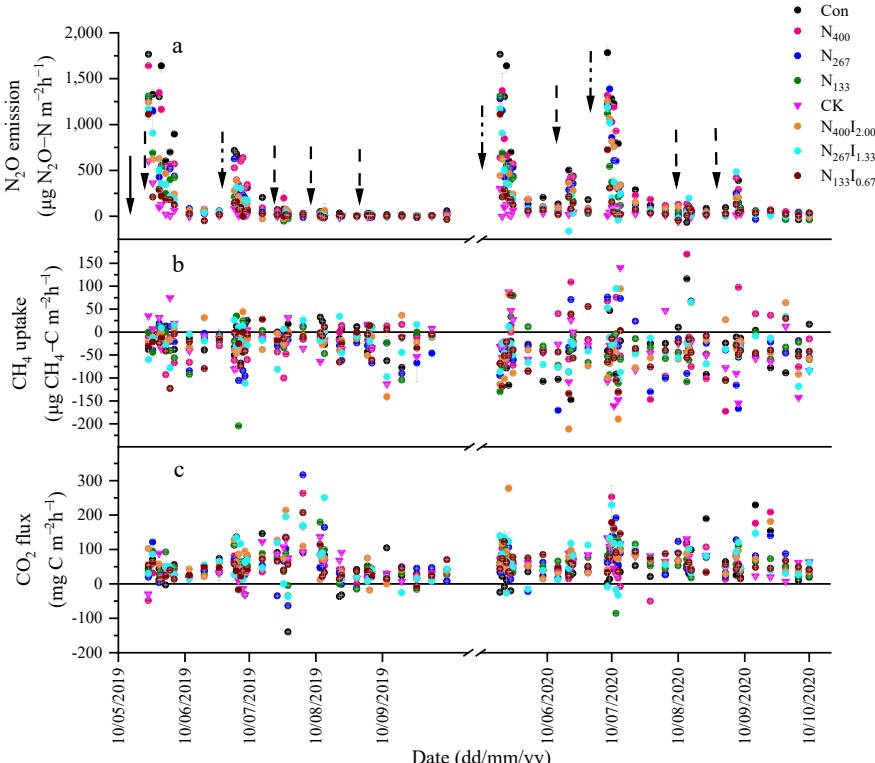

**Figure 3.** Dynamics of $N_2O$ emission (**a**), $CH_4$ uptake (**b**), and $CO_2$ flux (**c**) during the experimental period.

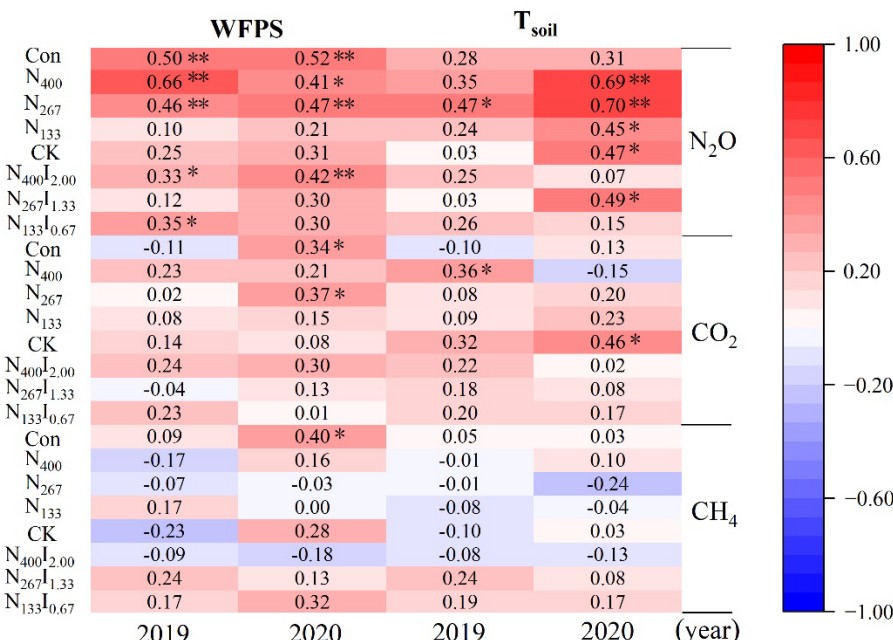

**Figure 4.** Pearson's correlation between GHG emissions and WFPS or $T_{soil}$. Con–$N_{133}I_{0.67}$ listed on the left represents different treatments. * and ** indicate correlation is significant at the level of 0.05 and 0.01, respectively.

Nitrogen fertilization significantly increased the cumulative $N_2O$ emissions compared with the CK treatment and the extent also varied with the addition of nitrapyrin and year ($p < 0.05$) (Table 1). When no nitrapyrin was added, the cumulative $N_2O$ emissions in 2019 and 2020 were significantly positively correlated with N application rate; the determination coefficients were 0.952 and 0.902, respectively (Figure 5). In contrast to Con treatment, the cumulative $N_2O$ emission of $N_{400}$ and CK decreased by 13.87–16.37%, 86.70–90.13%, respectively. Excluding the reference treatment, the cumulative $N_2O$ emissions were the highest when the N application rate was 400 kg·ha$^{-1}$ in 2019 and 2020, which were 5.72 kg·ha$^{-1}$ and 8.38 kg·ha$^{-1}$, respectively. Nitrapyrin addition significantly reduced the cumulative $N_2O$ emissions and $EF_{N2O}$ under the same N application rate. Compared with $N_{400}$, $N_{267}$ and $N_{133}$ treatments, $N_{400}I_{2.00}$, $N_{267}I_{1.33}$ and $N_{133}I_{0.67}$ treatments with nitrapyrin decreased cumulative $N_2O$ emissions by 45.1%, 26.6% and 45.9% in 2019 and 28.5%, 29.9% and 33.8% in 2020. In two years, in contrast with $N_{400}$, $N_{267}$ and $N_{133}$ treatments, the $EF_{N2O}$ of $N_{400}I_{2.00}$–$N_{133}I_{0.67}$ treatments were significantly lower than $N_{400}$, $N_{267}$ and $N_{133}$ treatments, which was significantly decreased by 53.3–69.6% in 2019 and 32.3–42.6% in 2020.

Table 1. Cumulative emissions of GHG, wolfberry yields, GWP and GHGI from wolfberry orchard with different treatments.

| Year | Treatment | | Yield (Mg·ha⁻¹) | Net Income (USD·ha⁻¹) | N₂O Cumulative Emission (kg·ha⁻¹) | N₂O EF$_{N2O}$ (%) | CH₄ Cumulative Emission (kg·ha⁻¹) | CO₂ Cumulative Emission (Mg·ha⁻¹) | GWP (kg·ha⁻¹) | GHGI (CO₂-eq kg ·Mg⁻¹) |
|---|---|---|---|---|---|---|---|---|---|---|
| 2019 | Reference treatment | Con | 7.20 ± 0.24 [cd] | 37,184 ± 1273 [c] | 6.84 ± 0.01 [a] | 0.89 ± 0.01 [d] | −0.71 ± 0.05 [a] | 1.67 ± 0.05 [a] | 2822.18 ± 6.48 [a] | 391.67 ± 4.27 [a] |
| | | CK | 6.37 ± 0.23 [e] | 33,197 ± 1236 [d] | 0.91 ± 0.06 [g] | - | −0.78 ± 0.03 [b] | 1.66 ± 0.05 [a] | 347.98 ± 25.43 [g] | 59.79 ± 4.79 [g] |
| | N fertilizer without nitrapyrin | N$_{400}$ | 7.44 ± 0.23 [bc] | 38,614 ± 1237 [abc] | 5.72 ± 0.17 [b] | 1.20 ± 0.04 [b] | −0.69 ± 0.04 [a] | 1.70 ± 0.04 [a] | 2354.72 ± 70.25 [b] | 316.49 ± 11.99 [b] |
| | | N$_{267}$ | 7.34 ± 0.25 [cd] | 38,148 ± 1333 [bc] | 3.72 ± 0.20 [c] | 1.05 ± 0.07 [c] | −0.77 ± 0.08 [b] | 1.66 ± 0.08 [a] | 1518.52 ± 81.14 [c] | 206.88 ± 9.44 [c] |
| | | N$_{133}$ | 7.03 ± 0.32 [d] | 36,821 ± 1685 [c] | 2.70 ± 0.04 [e] | 1.35 ± 0.01 [a] | −0.70 ± 0.03 [ab] | 1.68 ± 0.04 [a] | 1097.36 ± 18.16 [e] | 156.10 ± 1.37 [d] |
| | N fertilizer with nitrapyrin | N$_{400}$I$_{2.00}$ | 7.87 ± 0.17 [a] | 40,841 ± 883 [a] | 3.14 ± 0.09 [d] | 0.56 ± 0.01 [e] | −0.75 ± 0.02 [ab] | 1.65 ± 0.05 [a] | 1279.59 ± 37.53 [d] | 162.59 ± 8.60 [d] |
| | | N$_{267}$I$_{1.33}$ | 7.70 ± 0.15 [ab] | 40,059 ± 810 [ab] | 2.73 ± 0.11 [e] | 0.46 ± 0.03 [f] | −0.69 ± 0.05 [a] | 1.70 ± 0.07 [a] | 1111.48 ± 46.69 [e] | 144.35 ± 8.78 [e] |
| | | N$_{133}$I$_{0.67}$ | 7.04 ± 0.25 [d] | 36,629 ± 1333 [c] | 1.46 ± 0.03 [f] | 0.41 ± 0.04 [f] | −0.76 ± 0.01 [ab] | 1.66 ± 0.04 [a] | 581.50 ± 10.52 [f] | 82.60 ± 0.96 [f] |
| 2020 | Reference treatment | Con | 7.93 ± 0.29 [bc] | 41,068 ± 1558 [bcd] | 9.73 ± 0.63 [a] | 1.31 ± 0.08 [e] | −1.40 ± 0.04 [ab] | 2.25 ± 0.03 [a] | 3997.62 ± 263.60 [a] | 504.11 ± 11.71 [a] |
| | | CK | 6.71 ± 0.34 [f] | 37,666 ± 853 [f] | 0.96 ± 0.13 [f] | - | −1.30 ± 0.13 [ab] | 2.15 ± 0.03 [a] | 351.27 ± 54.53 [f] | 52.35 ± 2.03 [h] |
| | N fertilizer without nitrapyrin | N$_{400}$ | 8.01 ± 0.12 [bc] | 41,658 ± 626 [bc] | 8.38 ± 0.18 [b] | 1.86 ± 0.02 [c] | −1.35 ± 0.10 [ab] | 2.22 ± 0.06 [a] | 3437.61 ± 78.44 [b] | 429.16 ± 4.10 [b] |
| | | N$_{267}$ | 7.73 ± 0.14 [cd] | 40,244 ± 732 [cde] | 6.22 ± 0.35 [c] | 1.97 ± 0.13 [b] | −1.39 ± 0.02 [ab] | 2.12 ± 0.06 [a] | 2537.43 ± 145.72 [c] | 328.26 ± 19.87 [c] |
| | | N$_{133}$ | 7.45 ± 0.09 [de] | 39,048 ± 498 [ef] | 4.64 ± 0.11 [d] | 2.77 ± 0.03 [a] | −1.42 ± 0.03 [ab] | 2.12 ± 0.04 [a] | 1877.44 ± 45.90 [d] | 252.01 ± 3.07 [e] |
| | N fertilizer with nitrapyrin | N$_{400}$I$_{2.00}$ | 8.46 ± 0.11 [a] | 43,995 ± 567 [a] | 5.99 ± 0.19 [c] | 1.26 ± 0.02 [e] | −1.38 ± 0.12 [ab] | 2.18 ± 0.04 [a] | 2442.51 ± 81.99 [c] | 288.71 ± 5.34 [d] |
| | | N$_{267}$I$_{1.33}$ | 8.08 ± 0.16 [bc] | 42,067 ± 768 [b] | 4.36 ± 0.14 [d] | 1.27 ± 0.01 [e] | −1.44 ± 0.12 [b] | 2.20 ± 0.08 [a] | 1760.92 ± 62.30 [d] | 217.94 ± 2.97 [f] |
| | | N$_{133}$I$_{0.67}$ | 7.58 ± 0.13 [d] | 39,546 ± 694 [de] | 3.07 ± 0.09 [e] | 1.59 ± 0.04 [d] | −1.28 ± 0.04 [a] | 2.17 ± 0.08 [a] | 1231.04 ± 40.68 [e] | 162.41 ± 2.53 [g] |
| *p*-value | Year | | * | * | * | * | * | * | * | * |
| | N rate | | * | * | * | * | NS | NS | * | * |
| | nitrapyrin | | * | * | * | * | NS | NS | * | * |
| | Year × N rate | | NS | NS | NS | NS | NS | NS | NS | NS |
| | Year × nitrapyrin | | NS | NS | NS | NS | NS | NS | NS | NS |
| | N rate × nitrapyrin | | NS | NS | NS | NS | NS | NS | NS | NS |
| | Year × nitrapyrin × N rate | | NS | NS | NS | NS | NS | NS | NS | NS |

Note: All values in the table are mean ± standard error (*n* = 3); different letters after the data in the same column indicate significant differences among different treatments in the same year at 0.05 level. * indicates *p* < 0.05, NS indicates no significant difference. Cost–benefit analysis included assessment of the total costs, income from fruit sales and net economic benefit [43]. The average price of wolfberry (dried fruit) —5.97 USD·kg⁻¹, triple superphosphate—328 USD·t⁻¹, organic fertilizer—179 USD·t⁻¹, urea—296 USD·t⁻¹, labor cost of harvest—0.67 USD·kg⁻¹.

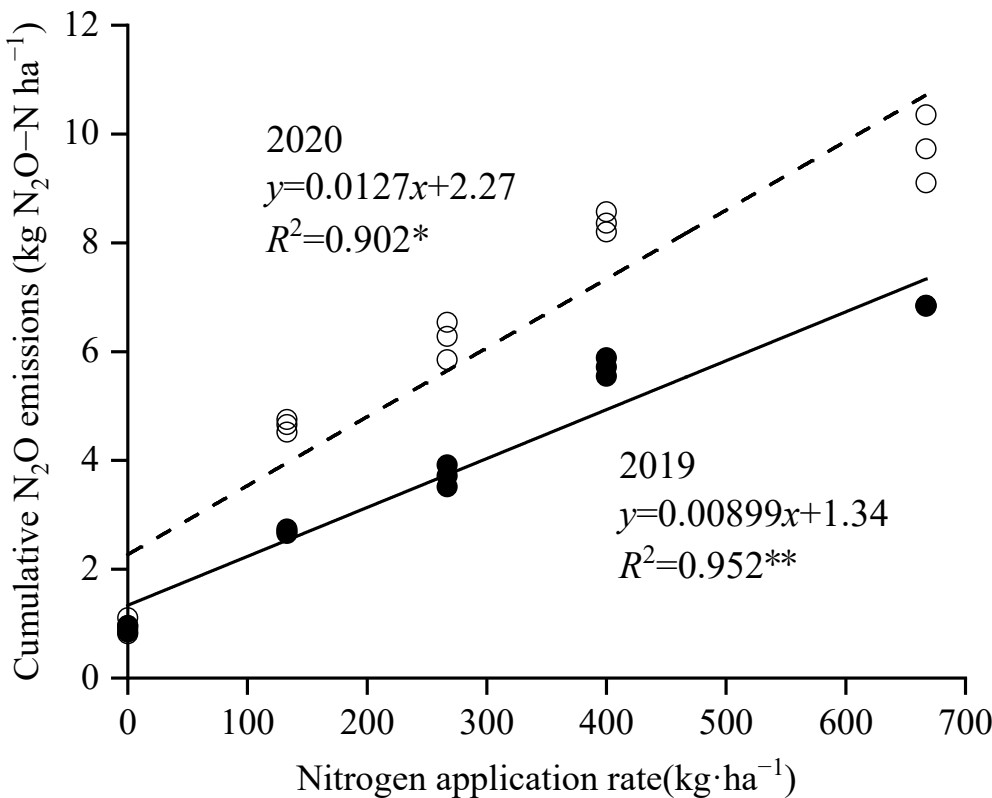

**Figure 5.** The response of cumulative emission of $N_2O$ on N rates. * and ** indicate correlation is significant at the level of 0.05 and 0.01, respectively.

### 3.2.2. $CH_4$ Uptake

$CH_4$ emissions were influenced by year ($p < 0.05$), but nitrapyrin and N fertilizers were not. Likewise, the combined effects of N application rate, nitrification inhibitors, and years did not affect $CH_4$ emissions (Table 1). The $CH_4$ emissions were negative during the growing season (Figure 3b). The highest values of $CH_4$ emission were 75 $\mu g \cdot m^{-2} \cdot h^{-1}$ and 170 $\mu g \cdot m^{-2} \cdot h^{-1}$, while absorption peaks were 205 $\mu g \cdot m^{-2} \cdot h^{-1}$ and 211 $\mu g \cdot m^{-2} \cdot h^{-1}$ in 2019 and 2020. The $CH_4$ cumulative emissions were not obviously different among treatments. Moreover, $CH_4$ emissions showed no significant correlation between soil moisture and 10 cm depth soil temperature (Figure 4).

### 3.2.3. $CO_2$ Flux

Similar to $CH_4$ emissions, the $CO_2$ flux was affected by year ($p < 0.05$) and the law of $CO_2$ flux was in line with the atmospheric variation with peaks in July and August (Figure 3c). $CO_2$ flux increased to 314 $mg \cdot m^{-2} \cdot h^{-1}$ from May to July in 2019, then slowly decreased to $-35$ $mg \cdot m^{-2} \cdot h^{-1}$ in early October. The peak value of $CO_2$ flux was 253 $mg \cdot m^{-2} \cdot h^{-1}$ in July and the minimum was 10 $mg \cdot m^{-2} \cdot h^{-1}$ at the beginning of October in 2020. In both years, cumulative $CO_2$ flux was not significantly different among treatments. During the growing season, $CO_2$ flux was positively correlated with soil temperature (Figure 4).

### 3.3. Wolfberry Yield, GWP and GHGI

Although the yield of wolfberry varied greatly among different years, the trend in yield variation among different treatments was consistent within both years. Besides, the yield was influenced by nitrapyrin and N application rates ($p < 0.05$) (Table 1). The yield ranged from 7.03–7.87 $Mg \cdot ha^{-1}$ and 7.58–8.46 $Mg \cdot ha^{-1}$ in 2019 and 2020, respectively. When no nitrapyrin was added, wolfberry yield increased first and then decreased with

N application rates increasing from 0 to 667 kg·ha$^{-1}$. When the N application rate was 400 kg·ha$^{-1}$, the yield of wolfberry reached the highest, which were 7.44 Mg·ha$^{-1}$ and 8.01 Mg·ha$^{-1}$ in 2019 and 2020, respectively. Compared with Con and CK treatment, the wolfberry yield of N$_{400}$ treatment increased by 1.01–3.33%, 16.80–19.37%, respectively. Meanwhile, the change in net income was similar with yield. At the same N application rate, the yield and net income were significantly higher in the treatment with nitrapyrin than that of without nitrapyrin, and N$_{400}$I$_{2.00}$ treatment produced the highest yield and net income in 2019 (7.87 Mg·ha$^{-1}$, 40,841 USD·ha$^{-1}$) and 2020 (8.46 Mg·ha$^{-1}$, 43,995 USD·ha$^{-1}$). Relative to N$_{400}$ treatment, N$_{400}$I$_{2.00}$ treatment had 5.7% and 5.6% higher yield in both years. When the combined N application rates were 423 kg·ha$^{-1}$ and 524 kg·ha$^{-1}$ in 2019 and 2020, the fruit net income was both the highest, 38,880 USD·ha$^{-1}$ and 41,483 USD·ha$^{-1}$, respectively (Figure 6).

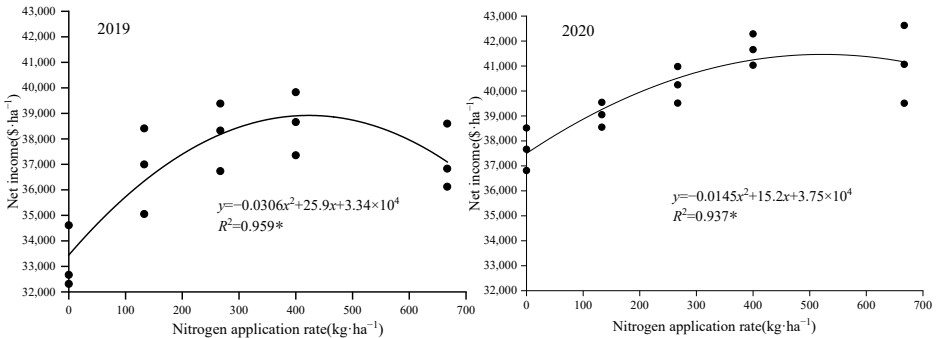

**Figure 6.** The response of net income of wolfberry on N rate in 2019 and 2020. * indicates correlation is significant at the level of 0.05.

Similar to the effect of yield, N application rates, nitrification inhibitors and different years can also affect GWP and GHGI independently ($p < 0.05$) while the two or three together did not (Table 1). GWP and GHGI had significant differences among treatments. N fertilizer input significantly improved GWP and GHGI, while the addition of nitrapyrin decreased GWP and GHGI under the same N application rates. The Con treatment produced the highest GWP of 2822 kg·ha$^{-1}$ and 3998 kg·ha$^{-1}$ in 2019 and 2020, respectively. In comparison to Con treatment, GWP of N$_{400}$, N$_{267}$ treatments declined by 16.6%, 46.2% and 14.0%, 36.5% in two years. Meanwhile, compared with N$_{400}$, N$_{267}$ and N$_{133}$ treatments, GWP of N$_{400}$I$_{2.00}$, N$_{267}$I$_{1.33}$ and N$_{133}$I$_{0.67}$ treatments decreased by 45.6%, 26.8% and 47.0%, while the GHGI decreased by 48.6%, 30.2% and 47.1%, respectively. GHGI ranged from 52.35–504.11 CO$_2$-eq Mg·kg$^{-1}$ in the study period. The GHGI peaked in Con treatment, with the lowest at N$_{133}$I$_{0.67}$ treatment. N$_{400}$I$_{2.00}$, N$_{267}$I$_{1.33}$ and N$_{133}$I$_{0.67}$ treatments lowered GHGI by 30.2–48.6% vs. N$_{400}$, N$_{267}$ and N$_{133}$ treatments without nitrapyrin.

## 4. Discussion

### 4.1. Effects of N Fertilizer Rate on Yield, GHG Emissions, GWP and GHGI

Scientific N management may reduce N fertilizer rate without compromising fruit yield [44,45]. Compared with no N fertilization, the input of N-fertilization can increase the yield of wolfberry; however, our study shows that excessive N application (e.g., Con) will reduce the yield of wolfberry. This may be due to the synergistic and antagonistic effects between nitrogen and other elements. High N input may lead to excess nutrient growth of wolfberry, causing unbalanced nutrient absorption [46]. Moreover, a previous study showed that long-term high N level led to reduced soil pH and root vigor [47]. In our study, compared to 667 kg·ha$^{-1}$ N treatment, a reduction of approximately 40% in N rate did not reduce wolfberry yield, which indicated that 667 kg·ha$^{-1}$ N was excessive for a one-year growth cycle of wolfberry. Therefore, on the basis of maintaining or increasing the yield, appropriate N amount and timing can improve economic benefits for the sustainable development of wolfberry industry. In combination with economic

benefits, when no nitrification inhibitor is applied, an N application rate of 423 kg·ha$^{-1}$ in 2018 and 524 kg·ha$^{-1}$ in 2019, which we recommend in wolfberry orchards with similar fertility, maximized net income, increasing net income by 1–5% but savign N fertilizer by 21–37%, respectively, compared to Con treatment. Therefore, excessive N input would reduce net income, while reducing N input reasonably could maximize net income and improve wolfberry production.

Zhang et al. [48] and Liu et al. [49] stated that N application in agriculture led to substantial GHG emissions in China, contributing nearly 50% of the total GHG emissions from agriculture in China. In our study, the main contribution of GWP was $N_2O$. The $N_2O$ emission was significantly positively correlated with nitrogen application rate ($p < 0.05$). $N_2O$ emission peaks occurred within one week after N application following irrigation. The occurrence of emission peaks typically coincided with high soil moisture (such as 19 May 2019 and 15 May 2020). One likely reason was that N input disrupted the C/N balance of microorganisms and stimulated microbial activity to some extent, which promoted $N_2O$ emission [50]. Moreover, N application provided sufficient substrates for microorganism, so that the intermediate products, $NH_2OH$ and $NO_2^-$ were released and improved in the process of soil ammonia oxidation, which was positively correlated with $N_2O$ emissions [51–53]. Another likely reason is high temperature improved the activity of soil microorganism and flood irrigation resulted in an anaerobic environment conducive to denitrification [54,55]. Denitrifying microorganisms under high temperature and humidity conditions increase $N_2O$ emissions. In this study, average daily $N_2O$ emissions from wolfberry receiving 0–667 kg·ha$^{-1}$ N ranged from 6.55–67.57 g $N_2O$-N ha$^{-1}$·d$^{-1}$. Similarly, in eastern China, average daily $N_2O$ emissions from an apple orchard were 71.78 g $N_2O$-N ha$^{-1}$·d$^{-1}$ [20]. At a similar nitrogen rate, we report lower $N_2O$ emissions. This may be due to the lower temperature and humidity on the Qinghai–Tibet Plateau, which reduces the $N_2O$ generation by inhibiting microorganisms to a certain extent. The soil N surplus due to high N application might increase $N_2O$ emissions; therefore, the cumulative emissions of $N_2O$ in 2020 was higher than that in 2019 in this study. Therefore, it is reasonable to expect that a reduction in the N fertilizer rate to an optimum level could substantially reduce $N_2O$ emissions.

Our study showed that N fertilizer application had no significant effect on $CH_4$ uptake. A meta-analysis showed that stimulation or inhibition of $CH_4$ oxidation depended on N additions, with a threshold of 100 kg·ha$^{-1}$ N [56]. However, the N application rate in our study was 133–667 kg·ha$^{-1}$, so the excessive N input might conceal the effect of inorganic N on methanotrophy. Moreover, $CH_4$ uptake is an extremely complex process, which is affected by environmental conditions and agricultural management [57]. Soil environment can affect soil respiration [46,58,59]. In our study, a significant positive correlation was detected between $CO_2$ flux and soil temperature. This might be due to the fact that $CO_2$ is the product of plant root and soil microbial respiration. High temperature stimulated this process and promoted $CO_2$ emissions. The temperature ranged from 5–25 °C in our study, and the increase in temperature might enhance the activity of soil microorganisms and accelerate the decomposition of organic matter, leading to an increase in $CO_2$ flux [60].

Increasing the N application rate greatly contributed to GHG emissions. Previous research indicated that N application increased annual $N_2O$ and $CH_4$ emissions, although it benefited atmospheric $CO_2$ sequestration into soil. The net effect is an increase in GWP and GHGI [61]. In our study, both GWP and GHGI of $N_{400}$ treatment were 40% lower compared with Con treatment, indicating that 667 kg·ha$^{-1}$ N was excessive for wolfberry production, causing high GHG emissions, low NUE and waste of agricultural resources. Excessive N fertilizer input is the main source of $N_2O$ emission and had a noticeable impact (increase) on GWP and GHGI. It is possible to reduce greenhouse gas emissions through optimizing N application rate, thereby further reducing the GWP of wolfberry orchard [62,63]. Results from this study also prove reasonable regulation on soil temperature and moisture was beneficial for mitigating GHG emissions.

### 4.2. Effects of Nitrapyrin on Yield, GHG Emissions, GWP and GHGI

Nitrification inhibitors have been used to regulate N conversion to nitrate-N and subsequent nitrous oxides. These inhibitors are also considered an effective measure to increase NUE. In our study, with the same N application rate, combining treatments with nitrapyrin increased yield. Because it inhibits the activity of ammonia-oxidizing bacteria, consequently delaying the conversion of $NH_4^+$ into $NO_3^-$, leading to the reduction in substrate that produces $N_2O$ emission [64]. Furthermore, the uptake of $NH_4^+$ required lower energy for plant N assimilation than $NO_3^-$ [65].

Our experiment showed the addition of nitrapyrin under the same N fertilizer application rate reduced the cumulative $N_2O$ emissions. The main reason was that nitrapyrin reduced the activity of soil nitrifying bacteria, and then inhibited the oxidation process of $NH_4^+$ to $NO_2^-$ in nitrification, thereby reducing the accumulation of $NO_3^-$-N and the emission of $N_2O$ [66]. Nitrification inhibitors have been widely reported as an effective management measure to reduce N loss and improve NUE, but the effect of nitrification inhibitors on $CO_2$ and $CH_4$ was rarely studied [67–69]. In our experiment, no significant difference was detected in net $CO_2$ and $CH_4$ between treatments. The effects of nitrification inhibitors on emissions may be jointly influenced by many factors, such as soil temperature, pH and soil oxidation–reduction potential. These factors were combined and lead to no significant effect of N fertilization on $CO_2$ and $CH_4$ emissions. The specific influencing process and interaction mechanism of these factors need further study.

GHGI reflected the total GHG emissions per unit of yield. This study showed that optimum N application combined with nitrapyrin significantly reduced GWP and GHGI, compared with N fertilizer alone. When $CH_4$ and $N_2O$ emissions were expressed as $CO_2$ equivalents, $N_2O$ was the main contributor to GWP. The contribution of $N_2O$ emission to GWP was as high as 99%. Nitrapyrin inhibited nitrification and reduced $N_2O$ emission, thereby reducing GWP and GHGI. These results showed that N application combined with nitrapyrin was a recommended cultivation method for high productivity and sustainable agriculture, considering environmental and economic efficiency.

### 4.3. Effects of N Fertilizer Rate Combined with Nitrapyrin on Yield, GHG Emissions, GWP and GHGI

Fertilization is an important means to maintain soil fertility and ensure stable food production. However, the phenomenon of excessive chemical fertilizer (especially N fertilizer) and low fertilizer utilization rate is widespread in crop production. Nitrification inhibitors are a widely used synergist for N fertilizer, effectively reducing N loss. In our study, 400 kg·ha$^{-1}$ N combined with 2.00 kg·ha$^{-1}$ nitrapyrin maximized net income in both years, which was the recommended N management. Furthermore, compared to 400 kg·ha$^{-1}$ N combined with 2.00 kg·ha$^{-1}$ nitrapyrin, 267 kg·ha$^{-1}$ N combined with 1.33 kg·ha$^{-1}$ nitrapyrin did not result in a great decrease in net income (by 2–4%), but that reduced GHG emission by 13–27%, and saved N fertilizer by 33%. Therefore, on the base of output and environmental impact assessment, 267 kg·ha$^{-1}$ N combined with 1.33 kg·ha$^{-1}$ nitrapyrin was the recommended N management in the study site. The combination helps to mitigate GHG emission and reduce N loss, thus improving wolfberry N uptake, and increasing NUE. An appropriate N fertilizer rate combined with nitrapyrin not only reduced GHG emissions, but also saved agricultural resources, which was beneficial for increasing economic benefits and achieving agricultural safe production. It was also the key to realizing the coordinated development of soil production and environmental function. If the optimized method was adopted across the Qaidam, the wolfberry yield would effectively increase, and N fertilizer input would decline by roughly 60%. Consequently, it was considered as a powerful strategy for the development of an agronomic system, benefiting both agriculture and environment.

## 5. Conclusions

Compared with Con treatment (i.e., standard current farmer practice), appropriately reducing N application rate could maintain wolfberry fruit yield and improve net income,

but greatly reduce GHG emissions and GWP. The recommended N application rate was 423 kg·ha$^{-1}$ in 2018 and 524 kg·ha$^{-1}$ in 2019, that maximized net income in wolfberry orchards with similar fertility. The nitrapyrin combined with N application could increase wolfberry fruit yield but reduce GHG emissions and N loss. Both management practices had positive effects, decreasing N$_2$O emission, while having no significant impact on CH$_4$ uptake and CO$_2$ emission. This strategy should be considered when the main concern is mitigating N$_2$O emission. The recommended N management was 400 kg·ha$^{-1}$ N combined with 2.00 kg·ha$^{-1}$ nitrapyrin maximized net income in the production of wolfberry in Qaidam. However, if simultaneously considering on economic profit and ecological benefit, 267 kg·ha$^{-1}$ N combined with 1.33 kg·ha$^{-1}$ was the optimal N application management that could maintain net income, but save N input and reduce N pollution.

**Author Contributions:** Conceptualization, H.S. and Y.G.; methodology, J.L. and H.S.; software, J.L. and Y.X.; validation, J.L. and Y.X.; formal analysis, J.L. and H.S.; investigation, H.S. and S.X.; resources, S.X. and R.Z.; data curation, J.L. and Y.X.; writing—original draft preparation, J.L. and J.M.; writing—review and editing, Y.X., Y.G. and H.S.; visualization, H.S. and J.L.; supervision, H.S., Y.X., and R.Z.; project administration, H.S.; funding acquisition, H.S and R.Z.; All authors have read and agreed to the published version of the manuscript.

**Funding:** This research was funded by Department of Science and Technology of Qinghai province, China, Grant No. [2020-HZ-805], and Open Fund of State Key Laboratory of Plateau Ecology and Agriculture of Qinghai University, Grant No. [2020-KF-001].

**Institutional Review Board Statement:** Not applicable.

**Informed Consent Statement:** Not applicable.

**Data Availability Statement:** All data in this study are available from the corresponding author.

**Conflicts of Interest:** The authors declare no conflict of interest.

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
