# Peer review of "Nitrogen Fertilizer and Nitrapyrin for Greenhouse Gas Reduction in Wolfberry Orchards on the Qinghai–Tibetan Plateau"

_agriculture, doi:10.3390/agriculture12071063_

Round 1
Reviewer 1 Report
The manuscript „N fertilizer and nitrapyrin for greenhouse gas reduction in wolfberry orchard on the Qinghai-Tibetan plateau“ focuses on the current problem caused by greenhouse gas emissions from nitrogen fertilizers. The study focuses on the use of a nitrification inhibitor (nitrapyrin) applied with nitrogen fertilizers to reduce N2O emission while considering Lycium barbarum yield.
My comments and recommendations are given below:
i. Lines 2: I recommend replacing the abbreviation “N” in the title of the manuscript “Nitrogen fertilizer ….”
ii. Abstract: unify the fertilizer or NI rate - somewhere it is with a dot (kg ha-1), somewhere without a dot (kg ha-1),
iii. keywords should not overlap with the title
iv. Line 46: Edit reference labels – wrong [7-8], right [7,8], the mislabelling also occurs on lines 57, 67 etc.
v. Line 48: wrong 49900, right 49,900
vi. Line 52: These references [13-15] are not related to the issue at hand!
vii. Does the use of nitrapyrine pose any ecological risks? “EPA has concluded that there are no ecological risks of concern when nitrapyrin is soil incorporated immediately after application.” It would be good to mention.
viii. Lines 95-98: Specify the methods (references) used in the determination of soil parameters.
ix. Line 102: Is the number of repetitions (3!) sufficient?
x. Line 101: The experiment was established in the same place in both years?
xi. Lines 103 – 106: Explain why the above nitrogen doses were used? Why 667 kg N ha-1? Do they relate to the needs of the plant for yield production?
xii. Line 108: Specify the composition of the fertilizer used. Was organic fertilizer applied in both years?
xiii. Lines 178-186: Statistical analysis deserves attention. Was normality test done? What about homogeneity of variance? Why not multivariate analysis? A two-way analysis is offered (year, fertilization treatment)? What method was used for follow-up testing? Tukey test or other?
xiv. Lines 189-197: I recommend including the part of chapter Environmental factors (Precipitation and air temperature during experiment) in the section Materials and Methods.
xv. Figure 2: Why does not the “fertilization scheme” the same in both years? I miss the black solid arrow in the second year.
xvi. Figure 2: What explains the high variability (SE values) in soil temperature measurements (especially in 2020)? Are the data and the related assessments (e.g., the correlations shown in Table 4) objective?
xvii. Figure 5: I recommend expressing cumulative N2O emission and yield in relation to N rate separately for treatments without and with inhibitor.
Reviewer 2 Report
This manuscript examines the effects of different treatment levels of nitrogen and nitrapyrin on wolfberry production in China's Qinghai Province. The authors present an interesting research topic suggesting that 400 kg N/ha combined with 2 kg nitrapyrin/ ha would be an optimal nitrogen rate for wolfberry production. Authors further state that this combination of nitrogen and netrapyrin would minimize the greenhouse gas intensity and global warming potential. I am sure that this study helps improve the understanding of nitrogen response to wolfberry yield. It is a well-written manuscript and has substantial merit. This manuscript gives insight into the importance of determining the optimal Nitrogen recommendation for maximizing wolfberry yield and at the same time protecting the environment. Whereas the topic itself is important, some issues warrant revision.
This study reported that the yield of wolfberry was significantly affected by different nitrogen level applications. However, the optimal level of fertilizer depends on several parameters such as residual soil nitrate-nitrogen and choice of the crop response function. Past studies have shown that crops get nitrogen from applied and carryover nitrogen from the previous year. The accumulation of carryover N significantly affects crop yield. Without accounting for carryover using the soil testing information before nitrogen application, residual nitrogen in crop production may not be efficient, and recommended nitrogen levels may be sub-optimal (under or over). Soil testing and fertilizer recommendation could also be an appropriate approach to understanding the optimal nitrogen rate for agricultural crop production. Authors should acknowledge this limitation or recommendation for future studies. Please refer to the following two recent articles.
- Dhakal, C., Lange, K., Parajulee, M. N., & Segarra, E. (2019). Dynamic optimization of nitrogen in plateau cotton yield functions with nitrogen carryover considerations. Journal of Agricultural and Applied Economics, 51(3), 385-401.
- Maaz, T., & Pan, W. (2017). Residual fertilizer, crop sequence, and water availability impact rotational nitrogen balances. Agronomy Journal, 109(6), 2839-2862.
Another neglected issue is fitting the appropriate crop yield response function as a fitting simple linear function, or ANOVA might not result in the optimal nitrogen doses. Please see the following manuscript discussing both of these issues in optimal fertilizer decision rules. I understand the scope of this manuscript is not about finding the best functional forms, but I would like to see mentioning this factor somewhere in the text.
- Dhakal, C., & Lange, K. (2021). Crop yield response functions in nutrient application: A review. Agronomy Journal, 113(6), 5222-5234.
To sum up, I agree with the authors that best nitrogen management practices would help reduce the negative consequences of over-application of nitrogen, but finding the best approach is still an ongoing debate. Several methods have been prosed that should be acknowledged.
Authors could discuss the above-mentioned issues in the introduction section to motivate the research question or in the discussion to provide the limitations of this study.
Be precise on the amount of nitrogen you recommend based on this study.
Please cite and provide references following the journal's format.
Round 2
Reviewer 1 Report
Dear authors,
Despite the revisions made to your manuscript, which I applaud, I still have comments, which I list below:
xii. Line 108: Specify the composition of the fertilizer used. Was organic fertilizer applied in both years?
Reply: The organic fertilizers used in the experiments of this study are the products of sheep manure fermentation and decomposing. The total organic matter content is 45%, and the N+P2O5+K2O content is 5%, which we have explained in the text. In order to keep experiment uniformity, our organic fertilizer application rate and method were same in both years.
In the case of fertiliser composition, specify the content of the individual nutrients separately (especially nitrogen), not in the NPK total!
xiii. Lines 178-186: Statistical analysis deserves attention. Was normality test done? What about homogeneity of variance? Why not multivariate analysis? A two-way analysis is offered (year, fertilization treatment)? What method was used for follow-up testing? Tukey test or other?
Reply: Through the one-way analysis of variance in different years, it can be clearly seen that the N application rate and the combined application of nitrapyrin have a significant impact on the observed indicators of the wolfberry experiment. The experiment included the two factors (N rate and nitrapyrin), but the rates of nitrapyrin was set as 0% and 0.5%. In fact, the rate nitrapyrin of 0% was taken as check treatment, so a multivariate analysis could not help here. If we set the rates of nitrapyrin as 0%, 0.5%, 1.0% and more, we would use a multivariate analysis. At the same time, we have added relevant content in table 1 for the purpose of further clarifying the effects of N application rate and nitrapyrin on various indicators. Tukey test was employed to detect the difference between treatments.
Apparently, you misunderstood my question. If you are evaluating the effect of fertilization in a multi-year experiment, it can be evaluated by a two-way analysis of variance, with the year (1) and fertilization (2) factor. In your case, you evaluated the effect of fertilization separately for each year. I wonder if the statistics will show the effect of fertilization in the average of both years.
Add information on the normality, homogeneity and follow-up tests performed to the manuscript (to 2.6. Statistical analysis)!
xvii. Figure 5: I recommend expressing cumulative N2O emission and yield in relation to N rate separately for treatments without and with inhibitor.
Reply: This is really a great suggestion. We have revised the result, conclusion and conclusion of the article according to your comments.
Instead of an economic evaluation of the effect of fertilization (N fertilizer rate), I expected the Cumulative N2O emissions and Wolfberry yield to be expressed separately for the: 1) N fertilizer without inhibitor and 2) N fertilizer with inhibitor.
Reviewer 2 Report
No comments
Author Response
The reviewer did not make any comments, and we have tried our best to revise the possible irregularities in the full text, such as spelling, capitalization, etc.